# Characterization and Phylogenetic Analysis of MADS-Box Gene Family in Magnoliids: Insights into the Evolution of Floral Morphogenesis in Angiosperms

**DOI:** 10.3390/plants14192991

**Published:** 2025-09-27

**Authors:** Haowei Chen, Haoyue Qu, Junmei Zhou, Junjie Pan, Zhoutao Wang, Liangsheng Zhang, Xiuxiu Li, Kejun Cheng

**Affiliations:** 1Fujian Universities Key Laboratory for Plant-Microbe Interaction, College of JunCao Science and Ecology (College of Carbon Neutrality), Fujian Agriculture and Forestry University, Fuzhou 350002, China; chenhaowei920423@163.com (H.C.); quhaoyue97@163.com (H.Q.); 2Chemical Biology Center, Lishui Institute of Agriculture and Forestry Sciences, Lishui 323000, China; zhoujunmei1008@163.com (J.Z.); junjpan@163.com (J.P.); 3School of Pharmaceutical Sciences, Fuchun Campus, Zhejiang Chinese Medical University, Hangzhou 310053, China; wangzt@zcmu.edu.cn; 4Genomics and Genetic Engineering Laboratory of Ornamental Plants, College of Agriculture and Biotechnology, Zhejiang University, Hangzhou 310058, China

**Keywords:** magnoliids, MADS-box genes, expression pattern, ABCDE model, evolution of floral development

## Abstract

Magnoliids represent one of the most basal lineages within angiosperms, and their ancestral floral morphology provides crucial insights into the evolution of flowers in angiosperms. MCM1-AGAMOUS-DEFICIENS-SRF (MADS)-box transcription factors play crucial roles in specifying floral organs. To understand their evolutionary history and functional divergence in magnoliids, we identified MADS-box genes, and conducted phylogenetic and expression analysis in 33 magnoliids and 8 other angiosperm plants. A total of 1310 MADS-box genes were identified and classified into Type I and Type II. The expansion of MADS-box genes in magnoliids mainly arose from whole-genome duplication events. In *Liriodendron chinensis* and *Chimonanthus praecox*, we identified floral homeotic MADS-box genes that are orthologous to the ABCDE model genes of floral organ identity determination. The broad expression pattern of A and B genes in floral organs and overlapping activity of ABCDE-model genes are consistent with the “shifting−fading borders” scheme proposed in basally diverging angiosperm lineages. Our results not only elucidate the driving forces underlying the diversification of MADS-box genes in magnoliids, but also shed light on the evolutionary models of floral development in angiosperms.

## 1. Introduction

MADS-box genes, named after four composite members, Minichromosome maintenance 1 (MCM 1) of *Saccharomyces cerevisiae*, AGAMOUS of *Arabidopsis thaliana*, DEFICIENS of *Antirrhinum majus*, and Serum response factor (SRF) of *Homo sapiens* [1], encode a family of transcription factors (TFs) that are fundamental regulators of plant development and floral morphogenesis in flowering plants [2]. Plant MADS-box genes can be classified into two super-clades known as Type I and Type II. Type I genes are further classified into the Mα, Mβ, and Mγ subgroups, characterized by a single MADS-box (M) domain and a variable C-terminal (C) domain. Type II MADS-box genes, encoding the M and C domains, as well as an intervening (I) domain and a keratin-like (K) domain, were classified into the MIKCC and MIKC* subgroups [3,4,5]. Numerous studies have shown that the divergence of major plant lineages, particularly flowering plants, is accompanied by the evolution of Type II MADS-box genes [6,7,8]. The MIKCC genes regulating floral morphogenesis were originally classified into three classes: A, B, and C genes [9]. Then, D class genes were found autonomously controlling the development of floral organs independently of A/B/C class gene activities, as reported in *Petunia hybrida* [10]. Finally, genetic analyses in *Arabidopsis thaliana* demonstrated that E genes are indispensable for specifying all four whorls when combined with A/B/C class factors, extending the classical ABC model into the more comprehensive ABCDE framework [11].

The magnoliids, including Canellales, Piperales, Laurales, and Magnoliales, comprising approximately ten thousand species, represent the third largest branch of the most diverse and species-rich angiosperms and are the pivotal phylogenetic lineages bridging all other extant angiosperms and gymnosperms [12]. Magnoliid plants retain ancestral floral features of early angiosperms and eudicots, including diversity in the number and arrangement of floral organs [13]. Despite the abundance of genome assemblies from eudicots and monocots, as well as newly sequenced genomes of several magnoliids, many questions about early mesangiosperm diversification and the evolution of floral organs remain uncertain [14,15,16,17,18,19,20,21].

Recently, several MADS-box genes in magnolia plants have been identified, such as AGL6 in *Chimonanthus salicifolius*, 63 MADS-box genes in the *Phoebe bournei* genome, and FUL-like genes in *Aristolochia fimbriata* [22,23,24]. The above studies primarily focused on the MADS-box genes within individual magnoliid plants while lacking comprehensive analyses of the evolution of MADS-box gene family and ABCDE floral organ model across the whole magnoliids. Evolutionary analysis of MADS-box genes across the whole magnoliids is essential for a deeper understanding of the origin of floral organs and will provide critical insights into how genetic innovations have shaped floral diversity in early-diverging angiosperms. To understand evolutionary history and functional divergence of MADS-box genes in magnoliids, in the present study, we sought to identify MADS-box family genes in magnoliids and other eight representative angiosperm plants, and then conducted phylogenetic and expression analysis.

## 2. Results

### 2.1. Identification of MADS-Box Genes in Magnoliids

We identified a total of 947 candidate MADS-box sequences in the genomes of 16 representative plant species that cover a broad diversity of plants, including two early angiosperms, three eudicots, three monocots, and eight magnoliids (Table 1 and Appendix A). In eudicots, *Arabidopsis thaliana* was found with the highest copy number (totaling 110) of MADS-box genes, closely followed by *Solanum lycopersicum* with 100 copies. Within monocots, *Oryza sativa* and *Sorghum bicolor* possess a similar number of MADS-box genes (64 and 74 copies each), which is nearly twice that of *Ananas comosus*. *Nymphaea colorata* and *Amborella trichopoda*, two early angiosperms, that were found containing 58 and 37 MADS-box genes, respectively. We found tremendous variation in MADS-box copy numbers among eight magnoliid plants in this study: the number per genome ranged from 31 to 85. Furthermore, a total of 363 MADS-box protein sequences were identified in 25 other magnoliid species using BLASTP in the OneKP transcriptome database, with most species exhibiting around a dozen copies (Table 1 and Appendix A). Finaly, a total of 1310 MADS-box genes were identified in this study (Appendix A).

Notably, except for certain magnoliids such as *Cinnamomum kanehirae*, *Chimonanthus praecox*, *Piper nigrum*, and *Liriodendron chinense*, most magnoliids possess less than 40 MADS-box genes, which is significantly lower than those found in early angiosperms, eudicots, and monocots (Appendix A). The number of MADS-box genes in *Cinnamomum kanehirae*, *Chimonanthus praecox*, *Piper nigrum*, and *Liriodendron chinense* is nearly twice that of other magnoliids, indicating a lineage-specific expansion of MADS-box genes in these taxa. Among them, *Cinnamomum kanehirae*, *Chimonanthus praecox*, and Liriodendron chinense exhibit significant expansion in the Type I group, whereas *Piper nigrum* shows predominant expansion in the Type II group.

### 2.2. Phylogenetic Analysis of MADS-Box Genes in Magnoliids

In plants, Type I and Type II MADS-box genes encode proteins with distinct conserved domains [25]. Based on a phylogenetic analysis, the Type I MADS-box genes were further classified into α, β, and γ subgroups (Figure 1a). The presence of Type I subgroups in monocots, eudicots and magnoliids suggests their wide phylogenetic distribution across plant categories. Based on domain variation and the known groups from *Arabidopsis thaliana* and *Nymphaea colorata*, Type II MADS-Box genes can be clustered into 15 distinct evolutionary clades: SEP, AGL16, AP1/FUL, FLC, AG, SOC1, SVP, AGL15, ANR1, TM8, AGL12, OsMADS32, AP3, BS, and MIKC* (Figure 1b).

The number of MIKCC-type gene copies varies widely among magnoliid plants, especially in the BS, AP3, SEP, ANR1, and SOC1 clades (Figure 2). For example, both in the AP3 and BS clades, *Piper nigrum* contains the highest number of members, with 11 members in AP3 and 7 members in the BS clade, whereas the other magnoliids possess only 2–5 and 1–2 members in AP3 and BS clades, respectively. In addition, most magnoliids generally show markedly fewer members in the SOC1 and AP1 clades than eudicots and monocots. Particularly, *Aristolochia fimbriata* without a WGD event was found to possess MIKCC-type gene copies comparable to those of the early-diverging angiosperm *Amborella trichopoda*.

### 2.3. The Evolutionary History and Expansion of ABCDE Model Genes in Magnoliids

To systematically investigate the evolutionary history of MIKCC-type MADS-box genes in magnoliids, with particular focus on the ABCDE model genes in flower development, we also identified MIKCC-type and ABCDE model genes from transcriptomic databases of 25 magnoliids in the OneKP database. Then, all identified ABCDE model genes in the clades of AP1/FUL, AG, AGL11, SEP, AGL6, AP3, PI, and BS from 8 currently assembled genomes and 25 transcriptomes of magnoliids, along with genes from other representative angiosperms, were used to construct a phylogenetic tree.

The phylogenetic tree of AP1/FUL genes shows clear lineage-specific diversification, with euAP1, euFUL, and FUL-like clades representing distinct lineages within angiosperms: monocots, eudicots, and magnoliids (left panel of Figure 3a). In magnoliids, AP1/FUL genes have duplicated to form three clades in Piperales and two clades in Laurales, suggesting gene duplication events in AP1/FUL genes (left panel of Figure 3a). During the evolution of angiosperms, AP1/FUL genes may have undergone at least four gene duplication events. Among them, two duplication events that occurred within the core eudicots, subsequently establishing three evolutionary types: euAP1, euFUL, and FUL-like. However, the euAP1/FUL clade, which mainly contains genes from monocots and eudicots, also includes a few genes from magnoliids, such as *Cinnamomum kanehirae* and *Persea americana* (right panel of Figure 3a). AP1/FUL genes from most magnoliid plants were also found to exhibit close evolutionary relationships with those in basal angiosperms (right panel of Figure 3a).

The AG subfamily was divided into two clades: AGL11 and AG. Unlike the AP1/FUL clades, each clade of the AG subfamily is composed of members from magnoliids, eudicots, and monocots (Figure 3b). This pattern indicates that an ancestral duplication event occurred in the AG subfamily after the divergence of angiosperms and gymnosperms, subsequently forming two distinct evolutionary lineages: the AG and AGL11. The AG genes from magnoliids represent an early-diverging lineage, clustering as sister clades to the AG genes from both monocots and eudicots (Figure 3b). Notably, the AGL11 genes are missing in many Piperaceae species, indicating a lineage-specific gene loss event of AGL11 genes in Piperaceae species.

The SEP subfamily may have undergone multiple gene duplication events during its evolution, with ancestral duplication likely occurring before the emergence of extant angiosperms, resulting in the formation of two distinct evolutionary lineages: AGL6 and SEP (Figure 4a). The *Aristolochia genus* retains only a single copy in each of the AGL6 and SEP lineages, whereas other magnoliids possess multiple copies owing to their unique gene duplication events.

A recent study has revealed that the AP3 subfamily also underwent a gene duplication event before the formation of extant angiosperms, thereby giving rise to two evolutionary lineages: AP3 and PI [26]. Therefore, the phylogenetic tree for the AP3 and BS lineages was reconstructed and classified into three clades: AP3, BS, and PI. The AP3/BS/PI genes in magnoliids display divergent topological distributions in different clades. In the AP3 and PI clades, magnoliids’ genes exhibit closer evolutionary relationships with those of eudicots, whereas in the BS clade, magnoliids’ genes show greater affinity to basal angiosperms (Figure 4b). We found dramatic expansion of MADS-box genes across these three clades in *Piper nigrum* and *Butyrospermum parkii*, with *Piper nigrum* exhibiting five members in both the AP3 and BS clades, while *Butyrospermum parkii* possess four copies in the PI clade. In the PI evolutionary lineage, we identified a shared gene duplication event in the Laurales and Magnoliales, which likely resulted from magnoliid-specific whole-genome duplication (WGD) event.

### 2.4. Expression Pattern Analysis of the Floral Homeotic MADS-Box Genes in Different Floral Organs

To further clarify the role of MADS-box genes in magnoliids’ flower developmental process, the ABCDE model genes of *Chimonanthus praecox* (*C. praecox*) and *Liriodendron chinensis* (*L. chinensis*) were identified, and their expression profiles in different floral organs were analyzed. A total of fifteen and fourteen orthologous floral homeotic genes were identified in *C. praecox* and *L. chinensis*, respectively. *C. praecox* was found to possess seven class A genes (from *Cs.AGL6a* to *Cs.AGL6f* and *Cs.AP1*), three class B genes (*Cs.AP3a*, *Cs.AP3b* and *Cs.AP3c*), three class C genes (*Cs.AGa*, *Cs.AGb* and *Cs.AGc*), and two class E genes (*Cs.SEPa* and *Cs.SEPb*). In *L. chinensis*, we identified four class A genes (*Lchi.AGLa*, *Lchi.AGLb*, *Lchi.AGLc* and *Lchi.AP1*), three class B genes (*Lchi.AP3a*, *Lchi.AP3b* and *Lchi.AP3c*), four class C genes (from *Lchi.AGa* to *Lchi.AGd*), and three class E genes (*Lchi.SEPa*, *Lchi.SEPb*, and *Lchi.SEPc*) (Figure 5). The expression profiling results of ABCDE model genes in two species showed the same two notable features: genes of different classes had different expression profiles in different floral organs, and genes of the same class had similar expression patterns.

In *Chimonanthus praecox*, the AGL6 genes of A class were specifically expressed conservatively at high levels in flower tissues (tepals, stamens, and pistils). *Cs.AGL6a*, *Cs.AGL6b*, *Cs.AGL6c*, *Cs.AGL6d* and *Cs.AGL6e* were expressed in tepals and stamens at a higher level compared to pistils, while *Cs.AGL6f* was expressed at a higher level in pistils than tepals and stamen. The three class B genes, *Cs.AP3a*, *Cs.AP3b* and *Cs.AP3c*, were highly expressed in tepals, stamens and pistils during the flowering stages, but were also expressed at lower levels in leaf, stem, root and bud. For genes of C class, *Cs.AGa* and *Cs.AGb* were found highly expressed in pistils, while *Cs.AGc* was expressed in both stamen and pistils (Figure 5a).

In *Liriodendron chinensis*, most Type I MADS-box genes were almost undetectable in all tissues, while Type II MADS-box genes of different clades have various expression patterns in different tissues (Appendix A). Among all floral organs, *Lchi.AP1* in A class exhibited a relatively low expression level, suggesting a potential attenuation of its functional activity in the A class model. However, the other three class A genes *Lchi.AGL6a*, *Lchi.AGL6b* and *Lchi.AGL6c* were mainly expressed in sepals and petals, and expressed at low levels in pistils and stamens, indicating an important role in A model function during floral development. Among three AP3 genes in class B, *Lchi.AP3b* and *Lchi.AP3c* were highly expressed in petals and stamens, and *Lchi.AP3a* was also highly expressed in sepals and pistils despite its relatively low overall expression levels. The four class C genes, from *Lchi.AGa* to *Lchi.AGd*, were mainly expressed in stamens and pistils, but almost not expressed in petals. Among the three SEP genes in class E, *Lchi.SEPc* was expressed in all examined flower organs, while *Lchi.SEPb* could be detected in sepals and petals, and *Lchi.SEPa* was mainly expressed in petals, stamens, and pistils (Figure 5b).

Therefore, we identified the floral homeotic MADS-box genes in *C. praecox* and *L. chinensis* and proposed a possible “ABCE” model for floral organ identity determination based on the expression pattern of those “ABCE” model genes (Figure 6b,c). In this model, the AGL6 and AP1 genes in class A determined sepals’ and petals’ identity; the AP3 genes controlled the B function and determined petal and stamen identity; classes C and D containing AG genes determined third stamen and carpel identity; and SEP genes included in class E determined fourth-whorl pistil identity. In addition, Figure 6a was constructed according to the published data [27].

## 3. Discussion

### 3.1. The MADS-Box Gene Families of Four Magnoliid Plants: Cinnamomum kanehirae, Chimonanthus praecox, Piper nigrum, and Liriodendron chinense Have Undergone Substantial Expansion, Which May Result from Whole-Genome Duplications (WGDs)

*Aristolochia fimbriata*, which has not undergone whole-genome duplications (WGDs), possesses the minimum number of MADS-box genes [28]. In contrast, *Chimonanthus praecox* and *Piper nigrum* have undergone two and three WGD events, respectively, and thus contain more MADS-box genes [29,30]. Some clades of MIKCc-type MADS-box genes are present in all phylogenetic lineages of the selected species, indicating that their origin mainly stems from gene duplication in ancestral angiosperms [7]. The shared presence of homologous AP1 genes in *Arabidopsis thaliana*, *Cinnamomum kanehirae*, and *Persea americana* suggests that magnoliids might share common gene duplication events with eudicots [31,32]. The same duplicated pattern was also observed in the AG subfamily, with a gene duplication event in early angiosperm ancestors forming two evolutionary lineages: AG and AGL11. Alternatively, a duplication event of the SEP subfamily was found to have occurred before the emergence of early angiosperms, implying that this gene duplication event is shared by monocots, eudicots, and magnoliids [33]. The observed conflict between the MADS-box gene trees and the species tree in magnoliids indicates possible incomplete lineage sorting (ILS) [34] during their evolution, a phenomenon particularly prevalent during the rapid diversification of magnoliid lineage. These results indicate a complex evolutionary history within this lineage.

### 3.2. The Expression Patterns of MADS-Box Genes in Magnoliids Exhibit Both Similarities to Those of Angiosperms and Lineage-Specific Characteristics Unique to Magnoliids

Type I MADS-box genes in *Liriodendron chinensis* and *Chimonanthus praecox* exhibit undetectable expression levels in floral organs (Appendix A), which is consistent with the expression patterns observed in angiosperms such as *Salvia miltiorrhiza*, *Arabidopsis thaliana*, and *Oryza sativa* [35,36,37]. In *Chimonanthus praecox* and *Liriodendron chinensis*, the expression levels of Type II MADS-box AP1 genes are lower, contrasting with the high-expression patterns observed in *Trioecious papaya*, *Crocus sativus*, and *Hordeum vulgare* [38,39,40]. However, the AGL6 genes exhibit high-expression levels in *Cinnamomum camphora*, a magnoliid plant from the Lauraceae family [31]. The AGL6 genes of magnoliids are mainly expressed in the first-whorl floral organs, which is similar to the expression pattern observed in the early flowering plant *Nymphaea colorata* (water lily) [41]. These results suggest that AGL6 might be an ancestral floral development regulator, playing a pivotal role in the morphogenesis of floral organs within basal angiosperms [41]. The AP3 genes were expressed across all floral organs, which is similar to the expression pattern of *Cinnamomum kanehirae* [27], but different from the typical expression patterns observed in eudicots [42].

### 3.3. The ABCDE Homologs in Magnoliids Generally Exhibited Broader Expression Ranges in Floral Organs

The B class genes AP3 of typical eudicots such as *Arabidopsis thaliana* exhibited a specific high expression level in petal and stamen (Figure 6a), whereas in *Liriodendron chinensis* and *Chimonanthus praecox*, these genes expressed broadly across all floral whorls (Figure 6b,c). The expanded expression range of B class genes may lead to homeotic transformations (e.g., sepal-to-petal conversions) or blur sepal–petal boundaries [43]. In addition, the co-expression of the A class gene AGL6 and the B class gene AP3 may maintain the ancestral state of floral organs in magnoliids by inhibiting tepal differentiation. For example, in *Chimonanthus praecox* and *Liriodendron chinensis*, completely undifferentiated and partially differentiated tepals can be observed, respectively. This is similar to *Nymphaea colorata*, representing an early-diverging lineage, where ABCDE homologs exhibit broader expression ranges and undifferentiated tepals [41]. This reveals a similarly extensive ancestral expression pattern for floral organ determination in early angiosperms. These results are also consistent with studies on oncidium and phalaenopsis in Orchidaceae, suggesting conserved regulatory mechanisms underlying tepal differentiation during the evolution of flower plants [44].

The broad and overlapping expression patterns of ABC/D genes in different floral organs of magnoliids indicate shifting boundaries of the ABCDE model for flower development [45]. Moreover, the expression levels of the C class genes in magnoliids gradually increase from the boundary to the center of floral organs, which suggests a fading border model in early angiosperms [46]. Based on these results, we propose that the ABCDE genes in magnoliid plants exhibit remarkable dynamicity and considerable flexibility in regulating floral morphogenesis, with their expression patterns simultaneously incorporating features of both the ‘shifting boundary’ and ‘fading border’ models. Therefore, the transition from the simplified floral organs of early angiosperms and magnoliids to the highly diversified floral organs of eudicots likely results from the functional diversification and conservation driven by WGDs, as well as increasingly distinct expression boundaries of ABCDE genes.

## 4. Materials and Methods

### 4.1. Data Collection

We collected 16 genome assemblies and corresponding annotations of recently published magnoliids and other representative angiosperms. The plant species in this study included early angiosperms, magnoliids, monocots, and eudicots. We also used multiple transcriptome sequences of magnoliids extracted from the OneKP project [47]. The genomic data sources, including whole-genome assemblies and transcriptomes, are comprehensively listed in Table 2.

### 4.2. Identification of MADS-Box Genes

MADS-box homologs were identified utilizing an HMMER [48] search based on the conserved functional domain of MADS-box (PF00319), with an e-value threshold of 1e-5. The MADS-box sequence (AT2G22540) from *Arabidopsis thaliana* was utilized as a query for online BLASTP [49] search in the OneKP database using the same e-value cut-off. Following the removal of redundant and incomplete sequences, the integrity of the MADS-box domain structures in the retrieved sequences was confirmed through validation with both the Pfam database and the SMART website [50,51].

### 4.3. Classification and Phylogenetic Analysis of MADS-Box Genes

All candidate MADS-box protein sequences were aligned using MAFFT v7.427 [52] with its default settings. Phylogenetic trees were efficiently and accurately constructed employing the JTT + CAT model with approximating maximum likelihood in FastTree [53]. The phylogenetic trees of the MADS-box subfamily were refined and visualized using MEGA7 [54]. Finally, phylogenetic trees were classified and annotated using the online platform EvolView [55].

### 4.4. Expression Analysis of MADS-Box Genes in Magnoliids Using RNA-Seq Data

RNA expression data for *Liriodendron chinensis* and *Chimonanthus praecox* were obtained from the Sequence Read Archive (SRA) of EBI (https://www.ebi.ac.uk/ (accessed on 26 September 2025)) and NCBI (https://www.ncbi.nlm.nih.gov/ (accessed on 26 September 2025)). The SRA accession numbers are PRJNA277997 and PRJNA600650, respectively. SRA data were converted to fastq format using the SRA Toolkit (v2.11.0). Reads were aligned to reference genomes using Hisat2 [56]. Gene expression levels were quantified as fragments per kilobase of exon per million fragments mapped (FPKM). Read counts were generated using FeatureCounts v1.6 [57], and differentially expressed genes were identified using DESeq2 v1.20.0 [58]. Their expression levels were computed using the Ballgown R package (v1.8.0) [56].

## 5. Conclusions

In this study, we identified the MADS-box gene family in magnoliids and analyzed their evolutionary history and expression patterns across angiosperms. The expansion in MADS-box genes in several magnoliid species was primarily driven by WGDs. The expression patterns of MADS-box genes in magnoliids reveal a particular ABCDE model with fading borders of gene expression and gradual transitions in organ identity. Shifting boundaries of ABCDE gene expression may have contributed to the diversification of the magnoliid flower from those of early angiosperms and some monocots. This study demonstrates the evolutionary history of the MADS-box gene family and provides novel insights into the mechanisms underlying the evolution of flowers in angiosperms.

## Figures and Tables

**Figure 1 plants-14-02991-f001:**
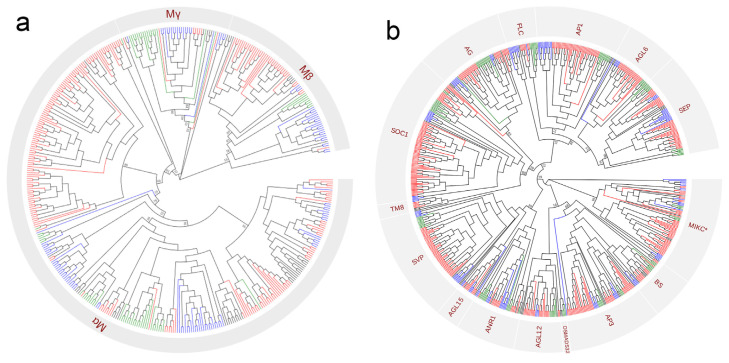
The evolutionary relationship and classification of Type I (**a**) and Type II (**b**) MADS-box genes in magnoliid and other angiosperm plants. MADS-box genes in early-diverging angiosperms, monocots, eudicots, and magnoliids are colored with black, green, blue, and red, respectively. A total of 1310 MADS-box genes were identified from genomic data and OneKP transcriptome database.

**Figure 2 plants-14-02991-f002:**
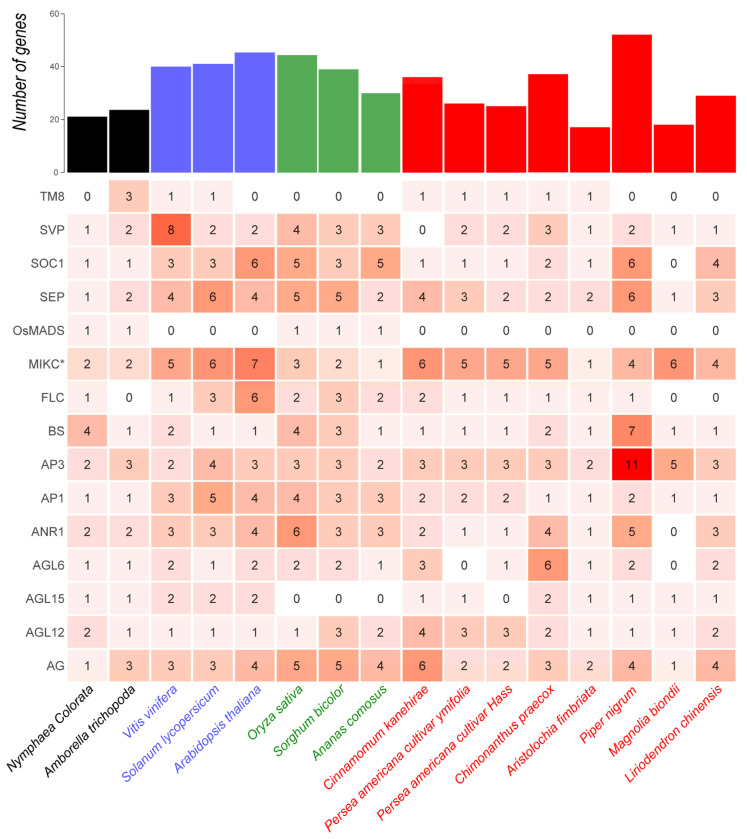
The detailed numbers of Type II MADS-box genes in magnoliids and other angiosperm plants. The total number of Type II MADS-box genes in each species is shown in the upper bar plot. In the heatmap, the red color gradient and values in each grid are directly proportional to gene counts. The early-diverging angiosperms, eudicots, monocots, and magnoliids are colored with black, blue, green, and red, respectively.

**Figure 3 plants-14-02991-f003:**
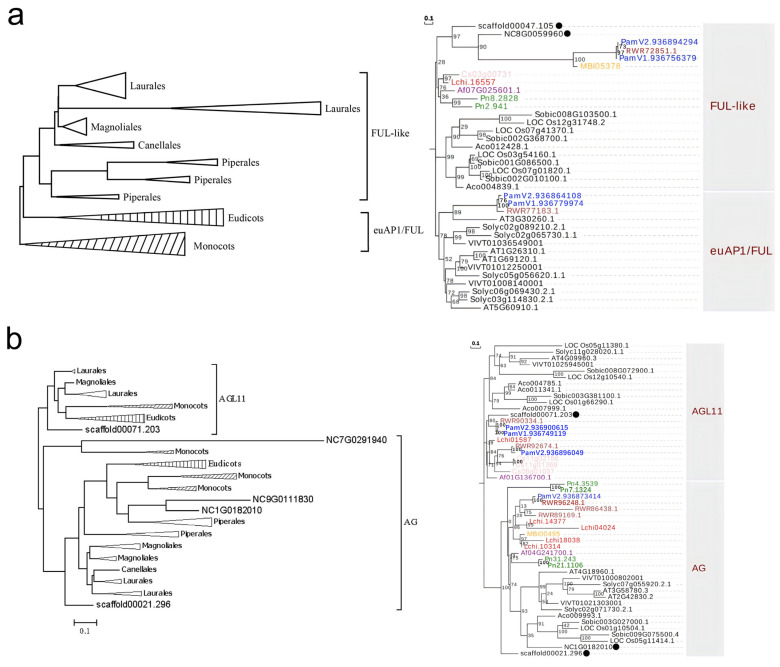
Phylogenetic relationships of AP1/FUL (**a**) and AG and AGL11 (**b**) MADS-box genes in angiosperms. In the left panel, genes from 8 currently assembled genomes and 25 transcriptomes of magnoliids, as well as other representative angiosperms, were used to construct simplified phylogenetic tree for each clade. The triangles with vertical and diagonal hatch shading represent monocots and eudicots, respectively. In the right panel, phylogenetic trees only include genes from 16 assembled genomes of magnoliids and other angiosperms. The genes of *Cinnamomum kanehirae, Aristolochia fimbriata, Magnolia biondii*, *Persea americana*, *Piper nigrum*, *Liriodendron chinensis*, and *Chimonanthus praecox* are colored brown, purple, yellow, blue, green, red, and pink, respectively. In addition, the genes of basal angiosperms are marked with black solid circles.

**Figure 4 plants-14-02991-f004:**
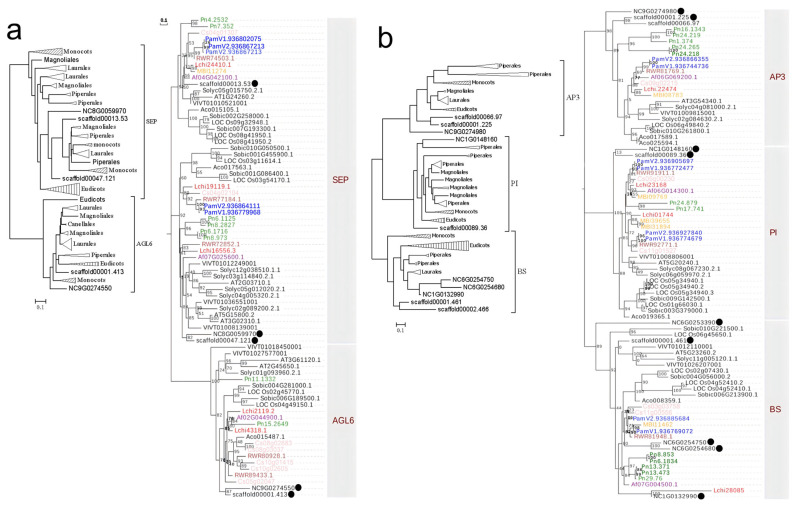
Phylogenetic relationships of SEP and AGL6 (**a**) and AP3 and PI and BS (**b**) MADS-box genes in angiosperms. All caption information refers to Figure 3.

**Figure 5 plants-14-02991-f005:**
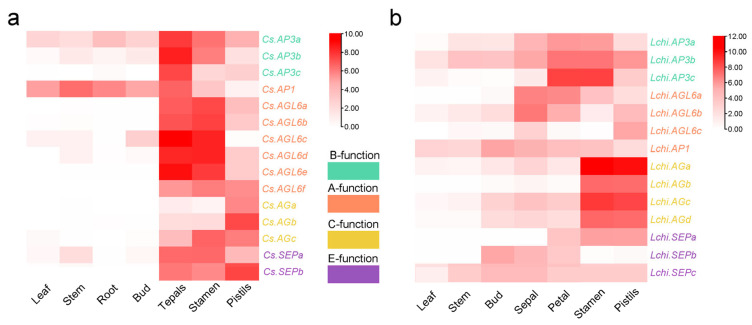
Expression profiling of ABCE model genes in *C. praecox* (**a**) and *L. chinensis* (**b**). The gene expression level was normalized as log2 (FPKM).

**Figure 6 plants-14-02991-f006:**
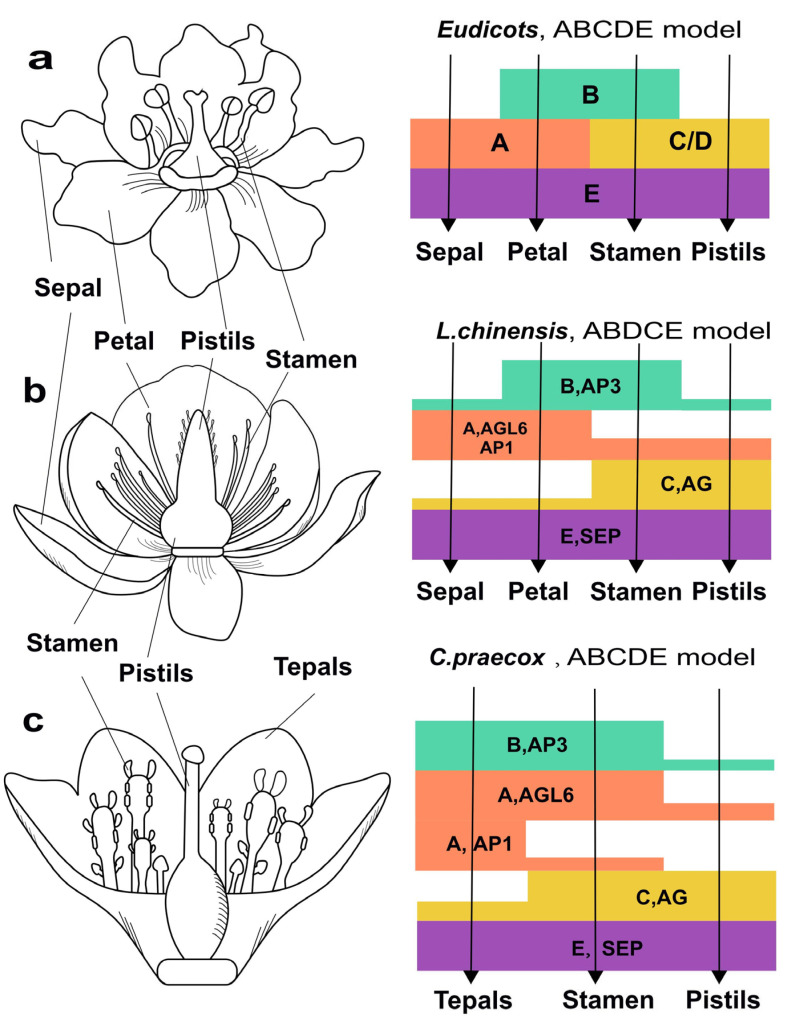
Expression patterns of floral homeotic MADS-box genes and ABCDE model of organ identity determination in eudicots (**a**), *L. chinensis* (**b**) and *C. praecox* (**c**).

**Table 1 plants-14-02991-t001:** Statistics of MADS-box genes in representative plant genomes.

Clade	Order	Family	Species	Abbr.	Type I	Type II	Total
Early-diverging Angiosperms	Nymphaeales	Nymphaeaceae	*Nymphaea colorata*	NC	37	21	58
	Amborellales	Amborellaceae	*Amborella trichopoda*	scaffold	13	24	37
Eudicots	Vitales	Vitaceae	*Vitis vinifera*	VIVT	10	40	50
	Solanales	Solanaceae	*Solanum lycopersicum*	Solyc	59	41	100
	Brassicales	Brassicaceae	*Arabidopsis thaliana*	AT	64	46	110
Monocots	Poales	Poaceae	*Oryza sativa*	LOC_Os	19	45	64
			*Sorghum bicolor*	Sobic	35	39	74
		Bromeliaceae	*Ananas comosus*	Aco	10	30	40
Magnoliids	Laurales	Lauraceae	*Cinnamomum kanehirae*	RWR	36	36	72
			*Persea americana cultivar drymifolia*	PamV1	7	26	33
			*Persea americana cultivar Hass*	PamV2	10	25	35
		Calycanthaceae	*Chimonanthus praecox*	Cs	48	37	85
	Piperales	Aristolochiaceae	*Aristo fimbriata*	Af	16	17	33
		Piperaceae	*Piper* *n* *igrum*	Pn	13	52	65
	Magnoliales	Magnoliaceae	*Magnolia biondii Pamp*	MBI	13	18	31
			*Liriodendron chinense*	Lchi	31	29	60
				Total	421	526	947

**Table 2 plants-14-02991-t002:** Genomic data sources used in this study.

Class	Species Name	Sources of Genomic Data
Early-diverging angiosperms	*Nymphaea colorata*	https://plants.ensembl.org/index.html (accessed on 26 September 2025)
	*Amborella trichopoda*	https://plants.ensembl.org/index.html (accessed on 26 September 2025)
Eudicots	*Vitis vinifera*	http://plants.ensembl.org/index.html (accessed on 26 September 2025)
	*Solanum lycopersicum*	https://phytozome.jgi.doe.gov/pz/portal.html (accessed on 26 September 2025)
	*Arabidopsis thaliana*	https://phytozome.jgi.doe.gov/pz/portal.html (accessed on 26 September 2025)
Monocots	*Oryza sativa*	https://phytozome.jgi.doe.gov/pz/portal.html (accessed on 26 September 2025)
	*Sorghum bicolor*	https://phytozome.jgi.doe.gov/pz/portal.html (accessed on 26 September 2025)
	*Ananas comosus*	https://phytozome.jgi.doe.gov/pz/portal.html (accessed on 26 September 2025)
Magnoliids	*Cinnamomum kanehirae*	https://www.ncbi.nlm.nih.gov/ (accessed on 26 September 2025)
	*Persea americana cultivar drymifolia*	https://genomevolution.org/ (accessed on 26 September 2025)
	*Persea americana cultivar Hass*	https://genomevolution.org/ (accessed on 26 September 2025)
	*Chimonanthus praecox*	https://www.ncbi.nlm.nih.gov/ (accessed on 26 September 2025)
	*Aristolochia fimbriata*	https://www.ncbi.nlm.nih.gov/ (accessed on 26 September 2025)
	*Piper nigrum*	https://www.ncbi.nlm.nih.gov/ (accessed on 26 September 2025)
	*Magnolia biondii Pamp.*	https://datadryad.org/ (accessed on 26 September 2025)
	*Liriodendron chinensis*	https://www.ebi.ac.uk/ (accessed on 26 September 2025)
	*Annona muricata*	https://db.cngb.org/onekp/ (accessed on 26 September 2025)
	*Eupomatia bennettii*	
	*Magnolia grandiflora*	
	*Michelia maudiae*	
	*Myristica fragrans*	
	*Uvaria microcarpa*	
	*Aristolochia elegans*	
	*Houttuynia cordata*	
	*Peperomia fraseri*	
	*Piper auritum*	
	*Saruma henryi*	
	*Saururus cernuus*	
	*Calycanthus floridus*	
	*Cassytha filiformis*	
	*Cinnamomum camphora*	
	*Gomortega keule*	
	*Gyrocarpus americanus*	
	*Idiospermum australiense*	
	*Laurelia sempervirens*	
	*Lindera benzoin*	
	*Persea borbonia*	
	*Peumus boldus*	
	*Sassafras albidum*	
	*Canella winterana*	
	*Drimys winteri*	

## Data Availability

The original contributions presented in this study are included in this article.

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
