# Peer review of "Characterization and Phylogenetic Analysis of MADS-Box Gene Family in Magnoliids: Insights into the Evolution of Floral Morphogenesis in Angiosperms"

_plants, 2025, doi:10.3390/plants14192991_

Round 1

Reviewer 1 Report

Comments and Suggestions for Authors

The manuscript by Chen and colleagues offers an interesting analysis of the evolutionary pattern of floral development genes in magnoliids. The authors used suitable methods and clearly presented their results. The discussion highlights the implications of the findings without unnecessary speculation. The conclusions are well-supported by the results and comparison with relevant literature. I have a few suggestions to improve the text.

1) The authors could clarify their objectives or questions at the end of the introduction.

2) They could avoid using the term "primitive" as a synonym for basal, ancestral, simple, or as an antonym for derived or complex.

3) In the methods section, some details and parameters could be included in the main text instead of just in the supplementary material.

4) Bootstrap support could be provided in trees, even if only for the main branches.

Reviewer 2 Report

Comments and Suggestions for Authors

Comments to the authors
This study sets out to explore the evolution of the key floral development genes in basal angiosperm plant lineages by identifying gene family members across multiple lineages and performing phylogenetic and expression analysis to help describe their evolutionary trajectories. The broad scope of this work will appeal widely to plant evolutionary biologists. The introduction provides good background on the MADS gene family and their roles in floral development. I suggest to further add the organs associated with each of the A/B/C genes for ease of reference. Gaps in knowledge about the evolution of flower development are argued well. This section assumes much prior taxonomic knowledge. Perhaps a short paragraph describing the different mentioned plant groups and their known relationships to each other would be helpful. The description of results for gene counts highlight interesting observations to help interpret gene copy change number. The description of results for phylogenies is more difficult to relate to the trees, partly because the trees are too small to read labels, and because basal angiosperms and eudicots do not have distinguishing colours. The expression results are summarised well and Figure 5 is useful to help interpret the ABCDE model. The discussion is organised well to highlight major interpretations of the results and to consider them in view of supporting literature. The results corroborate similar findings but this paper helps bring prior separate results into a larger overview of floral development evolution. Methods are described concisely but with sufficient information for repeatability. My main suggestions for improvement are to improve the presentation of trees so that the reader can more easily relate the descriptions to the figures. 

Specific comments
L21 The abstract refers to Magnoliids, while the title refers to Magnoliales.
L24-25 I suggest to add a line about searching for these genes too. Building a database of these sequences in an important study output too.
L40 Spell out MADS acronym at first use.
L59 confusing reference to multiple plant groups. Eudicots exclude Magnoliids.
L65 Give the full Latin name for the Phoebe plant mentioned here.
L78 Should Table S1 also be cited here?
L83 Add "that" between "angiosperms, were"
L88 Table S2 needs a column to indicate which samples were taken from the OneKP database.
L91 Replace "Nigrum" with "nigrum" here and elsewhere
L96-97 Write "Liriodendron chinense" in Italics
L110 Perhaps Figure 1a and 1b could be split into separate figures to allow them to be expand and better read clade labels.
L127 Is B-sister the same as BS? Be consistent with gene clade names. 
L149-150 The three gene clades in eudicots are not evident in Figure 3a 
L152-154 Tree tips belonging to basal angiosperms are not marked in Figure 3a.
L155 Again I suggest splitting these panels into different figures so they can be expanded for more readable labels. What does the hatch shading on left panel trees mean?
L161 Specify the colours used to mark each magnoliid species. 
L202-205 Reference Figure 4 here.
L217 Replace "Figure A1" with "Figure S1" What about Type I gene expression in Chimonanthus praecox?
L259-261 Revise, as this statement is contradictory as an early duplication event would lead to shared gene clades, not the opposite as stated.
L280 Add the Latin name of camphor tree for consistency.
Figure S1 needs an explanatory legend. 
